# GLOBAL-TO-LOCAL MEMORY POINTER NETWORKS FOR TASK-ORIENTED DIALOGUE

**Chien-Sheng Wu**[†*]**, Richard Socher**[‡] **& Caiming Xiong**[‡]
[‡]Salesforce Research
{rsocher,cxiong}@salesforce.com
[†]The Hong Kong University of Science and Technology
jason.wu@connect.ust.hk

## ABSTRACT

End-to-end task-oriented dialogue is challenging since knowledge bases are usually large, dynamic and hard to incorporate into a learning framework. We propose the global-to-local memory pointer (GLMP) networks to address this issue. In our model, a global memory encoder and a local memory decoder are proposed to share external knowledge. The encoder encodes dialogue history, modifies global contextual representation, and generates a global memory pointer. The decoder first generates a sketch response with unfilled slots. Next, it passes the global memory pointer to filter the external knowledge for relevant information, then instantiates the slots via the local memory pointers. We empirically show that our model can improve copy accuracy and mitigate the common out-of-vocabulary problem. As a result, GLMP is able to improve over the previous state-of-the-art models in both simulated bAbI Dialogue dataset and human-human Stanford Multi-domain Dialogue dataset on automatic and human evaluation.

## 1 INTRODUCTION

Task-oriented dialogue systems aim to achieve specific user goals such as restaurant reservation or navigation inquiry within a limited dialogue turns via natural language. Traditional pipeline solutions are composed of natural language understanding, dialogue management and natural language generation (Young et al., 2013; Wen et al., 2017), where each module is designed separately and expensively. In order to reduce human effort and scale up between domains, end-to-end dialogue systems, which input plain text and directly output system responses, have shown promising results based on recurrent neural networks (Zhao et al., 2017; Lei et al., 2018) and memory networks (Sukhbaatar et al., 2015). These approaches have the advantages that the dialogue states are latent without hand-crafted labels and eliminate the needs to model the dependencies between modules and interpret knowledge bases (KB) manually.

However, despite the improvement by modeling KB with memory network (Bordes & Weston, 2017; Madotto et al., 2018), end-to-end systems usually suffer from effectively incorporating external KB into the system response generation. The main reason is that a large, dynamic KB is equal to a noisy input and hard to encode and decode, which makes the generation unstable. Different from chit-chat scenario, this problem is especially harmful in task-oriented one, since the information in KB is usually the expected entities in the response. For example, in Table 1 the driver will expect to get the correct address to the gas station other than a random place such as a hospital. Therefore, pointer networks (Vinyals et al., 2015) or copy mechanism (Gu et al., 2016) is crucial to successfully generate system responses because directly copying essential words from the input source to the output not only reduces the generation difficulty, but it is also more like a human behavior. For example, in Table 1, when human want to reply others the *Valero*'s address, they will need to "copy" the information from the table to their response as well.

Therefore, in the paper, we propose the global-to-local memory pointer (GLMP) networks, which is composed of a global memory encoder, a local memory decoder, and a shared external knowledge. Unlike existing approaches with copy ability (Gulcehre et al., 2016; Gu et al., 2016; Eric &

---

[*]All work was done while the first author was an intern at Salesforce Research.

Table 1: An in-car assistant example on the navigation domain. The left part is the KB information and the right part is the conversation between a driver and our system.

| Point of interest (poi) | Distance | Traffic | Poi type | Address | | |
|---|---|---|---|---|---|---|
| Toms house | 3 miles | heavy | friend's house | 580 Van Ness Ave | Driver | I need gas |
| Coupa | 2 miles | moderate | coffee or tea place | 394 Van Ness Ave | System | GLMP: There is a gas station locally Valero is 4 miles away |
| | | | | | | Gold: Valero is 4 miles away |
| Panda express | 2 miles | no | Chinese restaurant | 842 Arrowhead Way | Driver | What is the address ? |
| Stanford express care | 5 miles | no | hospital | 214 El Camino Real | System | GLMP: Valero is located at 200 Alester Ave |
| Valero | 4 miles | heavy | gas station | 200 Alester Ave | | Gold: Valero is at 200 Alester Ave |
| Starbucks | 1 miles | heavy | coffee or tea place | 792 Bedoin St | Driver | Thank you! |

Manning, 2017; Madotto et al., 2018), which the only information passed to decoder is the encoder hidden states, our model shares the external knowledge and leverages the encoder and the external knowledge to learn a global memory pointer and global contextual representation. Global memory pointer modifies the external knowledge by softly filtering words that are not necessary for copying. Afterward, instead of generating system responses directly, the local memory decoder first uses a sketch RNN to obtain sketch responses without slot values but sketch tags, which can be considered as learning a latent dialogue management to generate dialogue action template. Then the decoder generates local memory pointers to copy words from external knowledge and instantiate sketch tags.

We empirically show that GLMP can achieve superior performance using the combination of global and local memory pointers. In simulated out-of-vocabulary (OOV) tasks in the bAbI dialogue dataset (Bordes & Weston, 2017), GLMP achieves 92.0% per-response accuracy and surpasses existing end-to-end approaches by 7.5% in full dialogue. In the human-human dialogue dataset (Eric et al., 2017), GLMP is able to surpass the previous state of the art on both automatic and human evaluation, which further confirms the effectiveness of our double pointers usage.

## 2 GLMP MODEL

Our model [1] is composed of three parts: global memory encoder, external knowledge, and local memory decoder, as shown in Figure 1(a). The dialogue history $X = (x_1, \ldots, x_n)$ and the KB information $B = (b_1, \ldots, b_l)$ are the input, and the system response $Y = (y_1, \ldots, y_m)$ is the expected output, where $n, l, m$ are the corresponding lengths. First, the global memory encoder uses a context RNN to encode dialogue history and writes its hidden states into the external knowledge. Then the last hidden state is used to read the external knowledge and generate the global memory pointer at the same time. On the other hand, during the decoding stage, the local memory decoder first generates sketch responses by a sketch RNN. Then the global memory pointer and the sketch RNN hidden state are passed to the external knowledge as a filter and a query. The local memory pointer returns from the external knowledge can copy text from the external knowledge to replace the sketch tags and obtain the final system response.

### 2.1 EXTERNAL KNOWLEDGE

Our external knowledge contains the global contextual representation that is shared with the encoder and the decoder. To incorporate external knowledge into a learning framework, end-to-end memory networks (MN) are used to store word-level information for both structural KB (KB memory) and temporal-dependent dialogue history (dialogue memory), as shown in Figure 1(b). In addition, the MN is well-known for its multiple hop reasoning ability (Sukhbaatar et al., 2015), which is appealing to strengthen copy mechanism.

**Global contextual representation.** In the KB memory module, each element $b_i \in B$ is represented in the triplet format as (Subject, Relation, Object) structure, which is a common format used to represent KB nodes (Miller et al., 2016; Eric et al., 2017). For example, the KB in the Table 1 will be denoted as {*(Tom's house, distance, 3 miles), ..., (Starbucks, address, 792 Bedoin St)*}. On the other hand, the dialogue context $X$ is stored in the dialogue memory module, where the speaker and temporal encoding are included as in Bordes & Weston (2017) like a triplet format. For instance, the first utterance from the driver in the Table 1 will be denoted as {*($user, turn1, I), ($user, turn1, need), ($user, turn1, gas)*}. For the two memory modules, a bag-of-word representation is used as the memory embeddings. During the inference time, we copy the object word once a memory position is pointed to, for example, *3 miles* will be copied if the triplet *(Toms house, distance, 3 miles)* is selected. We denote $Object(.)$ function as getting the object word from a triplet.

---

[1] https://github.com/salesforce

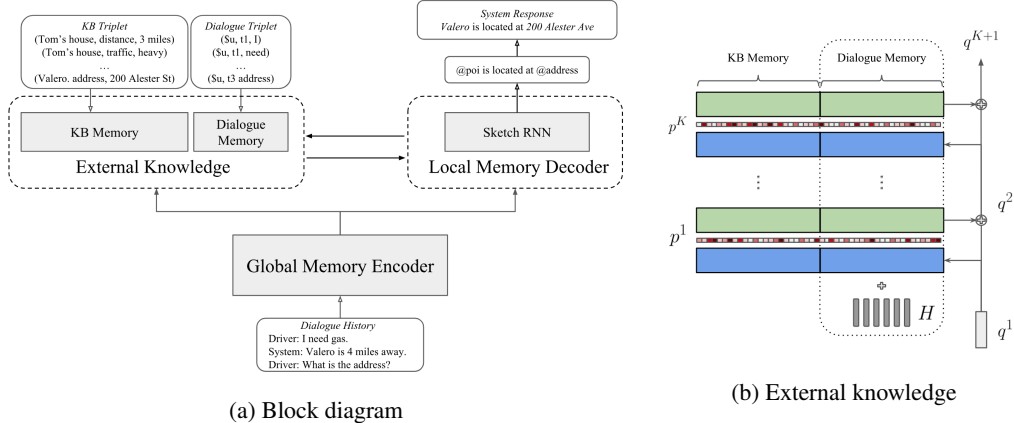

(a) Block diagram

(b) External knowledge

Figure 1: The proposed (a) global-to-local memory pointer networks for task-oriented dialogue systems and the (b) external knowledge architecture.

**Knowledge read and write.** Our external knowledge is composed of a set of trainable embedding matrices $C = (C^1, \ldots, C^{K+1})$, where $C^k \in \mathbb{R}^{|V| \times d_{emb}}$, $K$ is the maximum memory hop in the MN, $|V|$ is the vocabulary size and $d_{emb}$ is the embedding dimension. We denote memory in the external knowledge as $M = [B; X] = (m_1, \ldots, m_{n+l})$, where $m_i$ is one of the triplet components mentioned. To read the memory, the external knowledge needs a initial query vector $q^1$. Moreover, it can loop over $K$ hops and computes the attention weights at each hop $k$ using

$$p_i^k = \text{Softmax}((q^k)^T c_i^k), \tag{1}$$

where $c_i^k = B(C^k(m_i)) \in \mathbb{R}^{d_{emb}}$ is the embedding in $i^{th}$ memory position using the embedding matrix $C^k$, $q^k$ is the query vector for hop $k$, and $B(.)$ is the bag-of-word function. Note that $p^k \in \mathbb{R}^{n+l}$ is a soft memory attention that decides the memory relevance with respect to the query vector. Then, the model reads out the memory $o^k$ by the weighted sum over $c^{k+1}$ and update the query vector $q^{k+1}$. Formally,

$$o^k = \sum_i p_i^k c_i^{k+1}, \quad q^{k+1} = q^k + o^k. \tag{2}$$

## 2.2 Global Memory Encoder

In Figure 2(a), a context RNN is used to model the sequential dependency and encode the context $X$. Then the hidden states are written into the external knowledge as shown in Figure 1(b). Afterward, the last encoder hidden state serves as the query to read the external knowledge and get two outputs, the global memory pointer and the memory readout. Intuitively, since it is hard for MN architectures to model the dependencies between memories (Wu et al., 2018), which is a serious drawback especially in conversational related tasks, writing the hidden states to the external knowledge can provide sequential and contextualized information. With meaningful representation, our pointers can correctly copy out words from external knowledge, and the common OOV challenge can be mitigated. In addition, using the encoded dialogue context as a query can encourage our external knowledge to read out memory information related to the hidden dialogue states or user intention. Moreover, the global memory pointer that learns a global memory distribution is passed to the decoder along with the encoded dialogue history and KB information.

**Context RNN.** A bi-directional gated recurrent unit (GRU) (Chung et al., 2014) is used to encode dialogue history into the hidden states $H = (h_1^e, \ldots, h_1^e)$, and the last hidden state $h_n^e$ is used to query the external knowledge as the encoded dialogue history. In addition, the hidden states $H$ are written into the dialogue memory module in the external knowledge by summing up the original memory representation with the corresponding hidden states. In formula,

$$c_i^k = c_i^k + h_{m_i}^e \quad \text{if} \quad m_i \in X \text{ and } \forall k \in [1, K+1], \tag{3}$$

**Global memory pointer.** Global memory pointer $G = (g_1, \ldots, g_{n+l})$ is a vector containing real values between 0 and 1. Unlike conventional attention mechanism that all the weights sum to one,

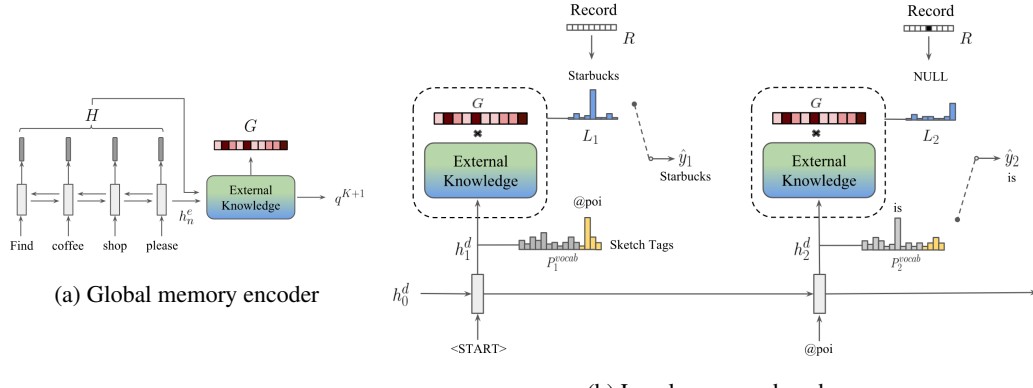

(a) Global memory encoder

(b) Local memory decoder

Figure 2: The proposed (a) global memory encoder and the (b) local memory decoder architecture.

each element in $G$ is an independent probability. We first query the external knowledge using $h_n^e$ until the last hop, and instead of applying the Softmax function as in (1), we perform an inner product followed by the Sigmoid function. The memory distribution we obtained is the global memory pointer $G$, which is passed to the decoder. To further strengthen the global pointing ability, we add an auxiliary loss to train the global memory pointer as a multi-label classification task. We show in the ablation study that adding this additional supervision does improve the performance. Lastly, the memory readout $q^{K+1}$ is used as the encoded KB information.

In the auxiliary task, we define the label $G^{label} = (g_1^l, \ldots, g_{n+l}^l)$ by checking whether the object words in the memory exists in the expected system response $Y$. Then the global memory pointer is trained using binary cross-entropy loss $Loss_g$ between $G$ and $G^{label}$. In formula,

$$g_i = \text{Sigmoid}((q^K)^T c_i^K), \quad g_i^l = \begin{cases} 1 & \text{if } Object(m_i) \in Y \\ 0 & \text{otherwise} \end{cases},$$

$$Loss_g = -\sum_{i=1}^{n+l}[g_i^l \times \log g_i + (1 - g_i^l) \times \log (1 - g_i)]. \tag{4}$$

### 2.3 LOCAL MEMORY DECODER

Given the encoded dialogue history $h_n^e$, the encoded KB information $q^{K+1}$, and the global memory pointer $G$, our local memory decoder first initializes its sketch RNN using the concatenation of $h_n^e$ and $q^{K+1}$, and generates a sketch response that excludes slot values but includes the sketch tags. For example, sketch RNN will generate "@*poi* is @*distance away*", instead of "*Starbucks is 1 mile away.*" At each decoding time step, the hidden state of the sketch RNN is used for two purposes: 1) predict the next token in vocabulary, which is the same as standard sequence-to-sequence (S2S) learning; 2) serve as the vector to query the external knowledge. If a sketch tag is generated, the global memory pointer is passed to the external knowledge, and the expected output word will be picked up from the local memory pointer. Otherwise, the output word is the word that generated by the sketch RNN. For example in Figure 2(b), a @poi tag is generated at the first time step, therefore, the word *Starbucks* is picked up from the local memory pointer as the system output word.

**Sketch RNN.** We use a GRU to generate a sketch response $Y^s = (y_1^s, \ldots, y_m^s)$ without real slot values. The sketch RNN learns to generate a dynamic dialogue action template based on the encoded dialogue ($h_n^e$) and KB information ($q^{K+1}$). At each decoding time step $t$, the sketch RNN hidden state $h_t^d$ and its output distribution $P_t^{vocab}$ are defined as

$$h_t^d = \text{GRU}(C^1(\hat{y}_{t-1}^s), h_{t-1}^d), \quad P_t^{vocab} = \text{Softmax}(W h_t^d) \tag{5}$$

We use the standard cross-entropy loss to train the sketch RNN, we define $Loss_v$ as.

$$Loss_v = \sum_{t=1}^{m} -\log(P_t^{vocab}(y_t^s)). \tag{6}$$

We replace the slot values in $Y$ into sketch tags based on the provided entity table. The sketch tags $ST$ are all the possible slot types that start with a special token, for example, @*address* stands for all the addresses and @*distance* stands for all the distance information.

**Local memory pointer.** Local memory pointer $L = (L_1, \ldots, L_m)$ contains a sequence of pointers. At each time step $t$, the global memory pointer $G$ first modify the global contextual representation using its attention weights,

$$c_i^k = c_i^k \times g_i, \quad \forall i \in [1, n+l] \text{ and } \forall k \in [1, K+1], \tag{7}$$

and then the sketch RNN hidden state $h_t^d$ queries the external knowledge. The memory attention in the last hop is the corresponding local memory pointer $L_t$, which is represented as the memory distribution at time step $t$. To train the local memory pointer, a supervision on top of the last hop memory attention in the external knowledge is added. We first define the position label of local memory pointer $L^{label}$ at the decoding time step $t$ as

$$L_t^{label} = \begin{cases} max(z) & \text{if } \exists z \text{ s.t. } y_t = Object(m_z), \\ n+l+1 & \text{otherwise.} \end{cases} \tag{8}$$

The position $n+l+1$ is a null token in the memory that allows us to calculate loss function even if $y_t$ does not exist in the external knowledge. Then, the loss between $L$ and $L^{label}$ is defined as

$$Loss_l = \sum_{t=1}^{m} -\log(L_t(L_t^{label})). \tag{9}$$

Furthermore, a record $R \in \mathbb{R}^{n+l}$ is utilized to prevent from copying same entities multiple times. All the elements in $R$ are initialized as 1 in the beginning. During the decoding stage, if a memory position has been pointed to, its corresponding position in $R$ will be masked out. During the inference time, $\hat{y}_t$ is defined as

$$\hat{y}_t = \begin{cases} \arg\max(P_t^{vocab}) & \text{if } \arg\max(P_t^{vocab}) \notin ST, \\ Object(m_{\arg\max(L_t \odot R)}) & \text{otherwise,} \end{cases} \tag{10}$$

where $\odot$ is the element-wise multiplication. Lastly, all the parameters are jointly trained by minimizing the weighted-sum of three losses ($\alpha, \beta, \gamma$ are hyper-parameters):

$$Loss = \alpha Loss_g + \beta Loss_v + \gamma Loss_l \tag{11}$$

## 3 EXPERIMENTS

### 3.1 DATASETS

We use two public multi-turn task-oriented dialogue datasets to evaluate our model: the bAbI dialogue (Bordes & Weston, 2017) and Stanford multi-domain dialogue (SMD) (Eric et al., 2017). The bAbI dialogue includes five simulated tasks in the restaurant domain. Task 1 to 4 are about calling API calls, modifying API calls, recommending options, and providing additional information, respectively. Task 5 is the union of tasks 1-4. There are two test sets for each task: one follows the same distribution as the training set and the other has OOV entity values. On the other hand, SMD is a human-human, multi-domain dialogue dataset. It has three distinct domains: calendar scheduling, weather information retrieval, and point-of-interest navigation. The key difference between these two datasets is, the former has longer dialogue turns but the regular user and system behaviors, the latter has few conversational turns but variant responses, and the KB information is much more complicated.

### 3.2 TRAINING DETAILS

The model is trained end-to-end using Adam optimizer (Kingma & Ba, 2015), and learning rate annealing starts from $1e^{-3}$ to $1e^{-4}$. The number of hop $K$ is set to 1,3,6 to compare the performance difference. The weights $\alpha, \beta, \gamma$ summing up the three losses are set to 1. All the embeddings are initialized randomly, and a simple greedy strategy is used without beam-search during the decoding stage. The hyper-parameters such as hidden size and dropout rate are tuned with grid-search over

Table 2: Per-response accuracy and completion rate (in the parentheses) on bAbI dialogues. GLMP achieves the least out-of-vocabulary performance drop. Baselines are reported from Query Reduction Network (Seo et al., 2017), End-to-end Memory Network (Bordes & Weston, 2017), Gated Memory Network (Liu & Perez, 2017), Point to Unknown Word (Gulcehre et al., 2016), and Memory-to-Sequence (Madotto et al., 2018).

| Task | QRN | MN | GMN | S2S+Attn | Ptr-Unk | Mem2Seq | GLMP K1 | GLMP K3 | GLMP K6 |
|------|-----|-----|-----|----------|---------|---------|---------|---------|---------|
| T1 | 99.4 (-) | 99.9 (99.6) | 100 (100) | 100 (100) | 100 (100) | 100 (100) | 100 (100) | 100 (100) | 100 (100) |
| T2 | 99.5 (-) | 100 (100) | 100 (100) | 100 (100) | 100 (100) | 100 (100) | 100 (100) | 100 (100) | 100 (100) |
| T3 | 74.8 (-) | 74.9 (2.0) | 74.9 (0) | 74.8 (0) | 85.1 (19.0) | 94.7 (62.1) | **96.3 (75.6)** | 96.0 (69.4) | 96.0 (68.7) |
| T4 | 57.2 (-) | 59.5 (3.0) | 57.2 (0) | 57.2 (0) | 100 (100) | 100 (100) | 100 (100) | 100 (100) | 100 (100) |
| T5 | **99.6 (-)** | 96.1 (49.4) | 96.3 (52.5) | 98.4 (87.3) | 99.4 (91.5) | 97.9 (69.6) | 99.2 (88.5) | 99.0 (86.5) | 99.2 (89.7) |
| T1 oov | 83.1 (-) | 72.3 (0) | 82.4 (0) | 81.7 (0) | 92.5 (54.7) | 94.0 (62.2) | **100 (100)** | **100 (100)** | 99.3 (95.9) |
| T2 oov | 78.9 (-) | 78.9 (0) | 78.9 (0) | 78.9 (0) | 83.2 (0) | 86.5 (12.4) | **100 (100)** | **100 (100)** | 99.4 (94.6) |
| T3 oov | 75.2 (-) | 74.4 (0) | 75.3 (0) | 75.3 (0) | 82.9 (13.4) | 90.3 (38.7) | 95.5 (65.7) | **96.7 (72.9)** | 95.9 (67.7) |
| T4 oov | 56.9 (-) | 57.6 (0) | 57.0 (0) | 57.0 (0) | 100 (100) | 100 (100) | 100 (100) | 100 (100) | 100 (100) |
| T5 oov | 67.8 (-) | 65.5 (0) | 66.7 (0) | 65.7 (0) | 73.6 (0) | 84.5 (2.3) | **92.0 (21.7)** | 91.0 (17.7) | 91.8 (21.4) |

Table 3: In SMD dataset, our model achieves highest BLEU score and entity F1 score over baselines, including previous state-of-the-art result from Madotto et al. (2018). (Models with * are reported from Eric et al. (2017), where the problem is simplified to the canonicalized forms.)

| | Automatic Evaluation | | | | | | | | |
|---|---|---|---|---|---|---|---|---|---|
| | Rule-Based* | KVR* | S2S | S2S + Attn | Ptr-Unk | Mem2Seq | GLMP K1 | GLMP K3 | GLMP K6 |
| BLEU | 6.6 | 13.2 | 8.4 | 9.3 | 8.3 | 12.6 | 13.83 | **14.79** | 12.37 |
| Entity F1 | 43.8 | 48.0 | 10.3 | 19.9 | 22.7 | 33.4 | 57.25 | **59.97** | 53.54 |
| Schedule F1 | 61.3 | 62.9 | 9.7 | 23.4 | 26.9 | 49.3 | 68.74 | **69.56** | 69.38 |
| Weather F1 | 39.5 | 47.0 | 14.1 | 25.6 | 26.7 | 32.8 | 60.87 | **62.58** | 55.89 |
| Navigation F1 | 40.4 | 41.3 | 7.0 | 10.8 | 14.9 | 20.0 | 48.62 | **52.98** | 43.08 |

| | Human Evaluation | | |
|---|---|---|---|
| | Mem2Seq | GLMP | Human |
| Appropriate | 3.89 | 4.15 | 4.6 |
| Humanlike | 3.80 | 4.02 | 4.54 |

the development set (per-response accuracy for bAbI Dialogue and BLEU score for the SMD). In addition, to increase model generalization and simulate OOV setting, we randomly mask a small number of input source tokens into an unknown token. The model is implemented in PyTorch and the hyper-parameters used for each task and the dataset statistics are reported in the Appendix.

## 3.3 RESULTS

**bAbI Dialogue.** In Table 2, we follow Bordes & Weston (2017) to compare the performance based on per-response accuracy and task-completion rate. Note that for utterance retrieval methods, such as QRN, MN, and GMN, cannot correctly recommend options (T3) and provide additional information (T4), and a poor generalization ability is observed in OOV setting, which has around 30% performance difference in Task 5. Although previous generation-based approaches (Ptr-Unk, Mem2Seq) have mitigated the gap by incorporating copy mechanism, the simplest cases such as generating and modifying API calls (T1, T2) still face a 6-17% OOV performance drop. On the other hand, GLMP achieves a highest 92.0% task-completion rate in full dialogue task and surpasses other baselines by a big margin especially in the OOV setting. No per-response accuracy loss for T1, T2, T4 using only the single hop, and only decreases 7-9% in task 5.

**Stanford Multi-domain Dialogue.** For human-human dialogue scenario, we follow previous dialogue works (Eric et al., 2017; Zhao et al., 2017; Madotto et al., 2018) to evaluate our system on two automatic evaluation metrics, BLEU and entity F1 score [2]. As shown in Table 3, GLMP achieves a highest 14.79 BLEU and 59.97% entity F1 score, which is a slight improvement in BLEU but a huge gain in entity F1. In fact, for unsupervised evaluation metrics in task-oriented dialogues, we argue that the entity F1 might be a more comprehensive evaluation metric than per-response accuracy or BLEU, as shown in Eric et al. (2017) that humans are able to choose the right entities but have very diversified responses. Note that the results of rule-based and KVR are not directly comparable because they simplified the task by mapping the expression of entities to a canonical form using named entity recognition and linking [3].

---

[2] BLEU: `multi-bleu.perl` script; Entity F1: Micro-average over responses.

[3] For example, they compared in "@*poi* is @*poi_distance away*," instead of "*Starbucks is 1_mile away.*"

Table 4: Ablation study using single hop model.

| | bAbI Dialogue OOV | | | | | SMD |
| | Per-response Accuracy | | | | | Entity F1 |
| | T1 | T2 | T3 | T4 | T5 | All |
| GLMP | 100 (-) | 100 (-) | 95.5 (-) | 100 (-) | 92.0 (-) | 57.25 (-) |
| GLMP w/o H | 90.4 (-9.6) | 85.6 (-14.4) | 95.4 (-0.1) | 100 (-0) | 86.2 (-5.3) | 47.96 (-9.29) |
| GLMP w/o G | 100 (-0) | 91.7 (-8.3) | 95.5 (-0) | 100 (-0) | 92.4 (+0.4) | 45.78 (-11.47) |

Moreover, human evaluation of the generated responses is reported. We compare our work with previous state-of-the-art model Mem2Seq [4] and the original dataset responses as well. We randomly select 200 different dialogue scenarios from the test set to evaluate three different responses. Amazon Mechanical Turk is used to evaluate system appropriateness and human-likeness on a scale from 1 to 5. As the results shown in Table 3, we see that GLMP outperforms Mem2Seq in both measures, which is coherent to previous observation. We also see that human performance on this assessment sets the upper bound on scores, as expected. More details about the human evaluation are reported in the Appendix.

**Ablation Study.**    The contributions of the global memory pointer $G$ and the memory writing of dialogue history $H$ are shown in Table 4. We compare the results using GLMP with $K = 1$ in bAbI OOV setting and SMD. GLMP without $H$ means that the context RNN in the global memory encoder does not write the hidden states into the external knowledge. As one can observe, our model without $H$ has 5.3% more loss in the full dialogue task. On the other hand, GLMP without $G$ means that we do not use the global memory pointer to modify the external knowledge, and an 11.47% entity F1 drop can be observed in SMD dataset. Note that a 0.4% increase can be observed in task 5, it suggests that the use of global memory pointer may impose too strong prior entity probability. Even if we only report one experiment in the table, this OOV generalization problem can be mitigated by increasing the dropout ratio during training.

**Visualization and Qualitative Evaluation.**    Analyzing the attention weights has been frequently used to interpret deep learning models. In Figure 3, we show the attention vector in the last hop for each generation time step. Y-axis is the external knowledge that we can copy, including the KB information and the dialogue history. Based on the question "*what is the address?*" asked by the driver in the last turn, the gold answer and our generated response are on the top, and the global memory pointer $G$ is shown in the left column. One can observe that in the right column, the final memory pointer successfully copy the entity *chevron* in step 0 and its address *783 Arcadia Pl* in step 3 to fill in the sketch utterance. On the other hand, the memory attention without global weighting is reported in the middle column. One can find that even if the attention weights focus on several point of interests and addresses in step 0 and step 3, the global memory pointer can mitigate the issue as expected. More dialogue visualization and generated results including several negative examples and error analysis are reported in the Appendix.

# 4 RELATED WORKS

**Task-oriented dialogue systems.** Machine learning based dialogue systems are mainly explored by following two different approaches: modularized and end-to-end. For the modularized systems (Williams & Young, 2007; Wen et al., 2017), a set of modules for natural language understanding (Young et al., 2013; Chen et al., 2016), dialogue state tracking (Lee & Stent, 2016; Zhong et al., 2018), dialogue management (Su et al., 2016), and natural language generation (Sharma et al., 2016) are used. These approaches achieve good stability via combining domain-specific knowledge and slot-filling techniques, but additional human labels are needed. On the other hand, end-to-end approaches have shown promising results recently. Some works view the task as a next utterance retrieval problem, for examples, recurrent entity networks share parameters between RNN (Wu et al., 2017), query reduction networks modify query between layers (Seo et al., 2017), and memory networks (Bordes & Weston, 2017; Liu & Perez, 2017; Wu et al., 2018) perform multi-hop design to strengthen reasoning ability. In addition, some approaches treat the task as a sequence generation problem. Lei et al. (2018) incorporates explicit dialogue states tracking into a delexicalized sequence generation. Serban et al. (2016); Zhao et al. (2017) use recurrent neural networks to generate final responses and achieve good results as well. Although it may increase the search space,

---

[4]Mem2Seq code is released and we achieve similar results stated in the original paper.

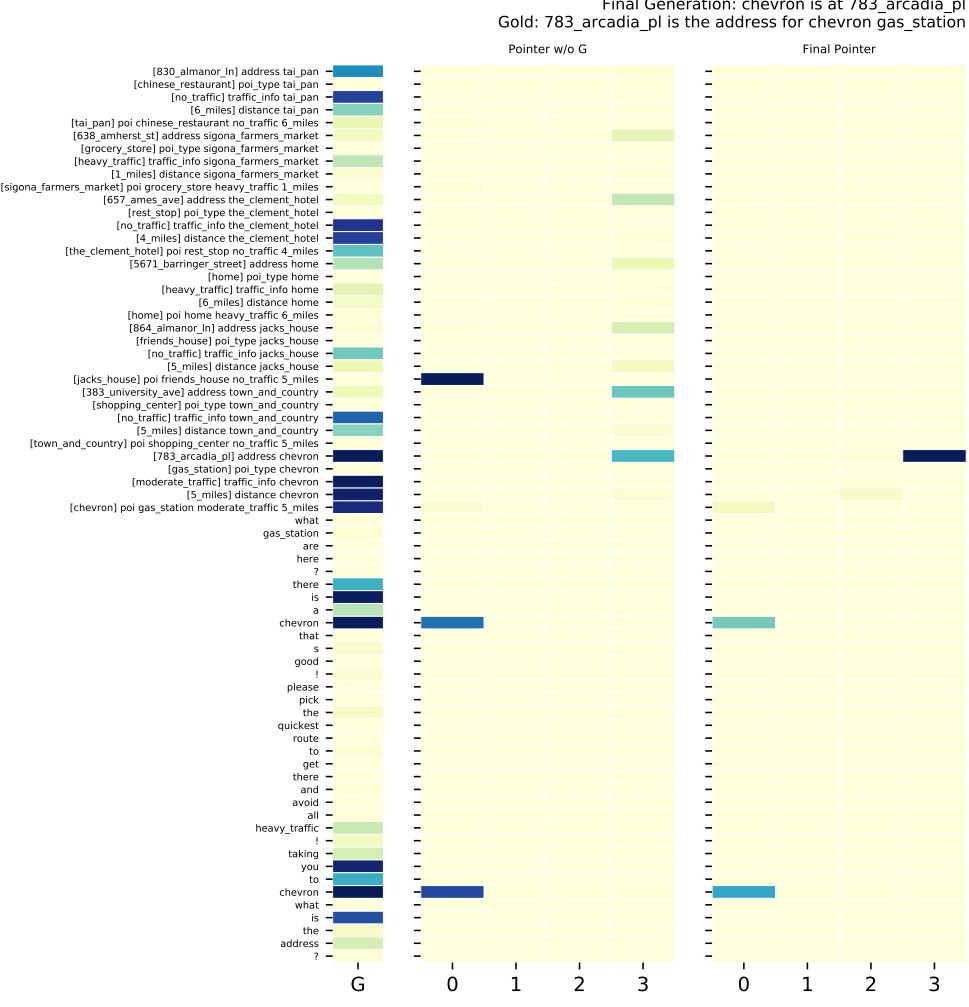

Figure 3: Memory attention visualization in the SMD navigation domain. Left column is the global memory pointer $G$, middle column is the memory pointer without global weighting, and the right column is the final memory pointer.

these approaches can encourage more flexible and diverse system responses by generating utterances token-by-token.

**Pointer network.** Vinyals et al. (2015) uses attention as a pointer to select a member of the input source as the output. Such copy mechanisms have also been used in other natural language processing tasks, such as question answering (Dehghani et al., 2017; He et al., 2017), neural machine translation (Gulcehre et al., 2016; Gu et al., 2016), language modeling (Merity et al., 2017), and text summarization (See et al., 2017). In task-oriented dialogue tasks, Eric & Manning (2017) first demonstrated the potential of the copy-augmented Seq2Seq model, which shows that generation-based methods with simple copy strategy can surpass retrieval-based ones. Later, Eric et al. (2017) augmented the vocabulary distribution by concatenating KB attention, which at the same time increases the output dimension. Recently, Madotto et al. (2018) combines end-to-end memory network into sequence generation, which shows that the multi-hop mechanism in MN can be utilized to improve copy attention. These models outperform utterance retrieval methods by copying relevant entities from the KBs.

**Others.** Zhao et al. (2017) proposes entity indexing and Wu et al. (2018) introduces recorded delexicalization to simplify the problem by record entity tables manually. In addition, our approach utilized recurrent structures to query external memory can be viewed as the memory controller

in Memory augmented neural networks (MANN) (Graves et al., 2014; 2016). Similarly, memory encoders have been used in neural machine translation (Wang et al., 2016) and meta-learning applications (Kaiser et al., 2017). However, different from other models that use a single matrix representation for reading and writing, GLMP leverages end-to-end memory networks to perform multiple hop attention, which is similar to the stacking self-attention strategy in the Transformer (Vaswani et al., 2017).

## 5 CONCLUSION

In the work, we present an end-to-end trainable model called global-to-local memory pointer networks for task-oriented dialogues. The global memory encoder and the local memory decoder are designed to incorporate the shared external knowledge into the learning framework. We empirically show that the global and the local memory pointer are able to effectively produce system responses even in the out-of-vocabulary scenario, and visualize how global memory pointer helps as well. As a result, our model achieves state-of-the-art results in both the simulated and the human-human dialogue datasets, and holds potential for extending to other tasks such as question answering and text summarization.

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

# A  TABLES

## A.1  TRAINING PARAMETERS

Table 5: Selected hyper-parameters in each dataset for different hops. The values is the embedding dimension and the GRU hidden size, and the values between parenthesis is the dropout rate. For all the models we used learning rate equal to 0.001, with a decay rate of 0.5.

|  | T1 | T2 | T3 | T4 | T5 | SMD |
|---|---|---|---|---|---|---|
| GLMP K1 | 64 (0.1) | 64 (0.3) | 64 (0.3) | 64 (0.7) | 128 (0.3) | 128 (0.2) |
| GLMP K3 | 64 (0.3) | 64 (0.3) | 64 (0.3) | 64 (0.7) | 128 (0.1) | 128 (0.2) |
| GLMP K6 | 64 (0.3) | 64 (0.3) | 64 (0.5) | 64 (0.5) | 128 (0.1) | 128 (0.3) |

## A.2  DATASET STATISTICS

Table 6: Dataset statistics for 2 datasets.

| Task | 1 | 2 | 3 | 4 | 5 | SMD | | |
|---|---|---|---|---|---|---|---|---|
|  |  |  |  |  |  | Calendar | Weather | Navigation |
| Avg. User turns | 4 | 6.5 | 6.4 | 3.5 | 12.9 | 2.6 | | |
| Avg. Sys turns | 6 | 9.5 | 9.9 | 3.5 | 18.4 | 2.6 | | |
| Avg. KB results | 0 | 0 | 24 | 7 | 23.7 | 66.1 | | |
| Avg. Sys words | 6.3 | 6.2 | 7.2 | 5.7 | 6.5 | 8.6 | | |
| Max. Sys words | 9 | 9 | 9 | 8 | 9 | 87 | | |
| Nb. Slot Types | 7 | | | | | 6 | 4 | 5 |
| Nb. Distinct Slot values | - | | | | | 79 | 65 | 140 |
| Vocabulary | 3747 | | | | | 1601 | | |
| Train dialogues | 1000 | | | | | 2425 | | |
| Val dialogues | 1000 | | | | | 302 | | |
| Test dialogues | 1000 + 1000 OOV | | | | | 304 | | |
| Total Nb. Dialogues | 4000 | 4000 | 4000 | 4000 | 4000 | 1034 | 997 | 1000 |

## A.3  HUMAN EVALUATION

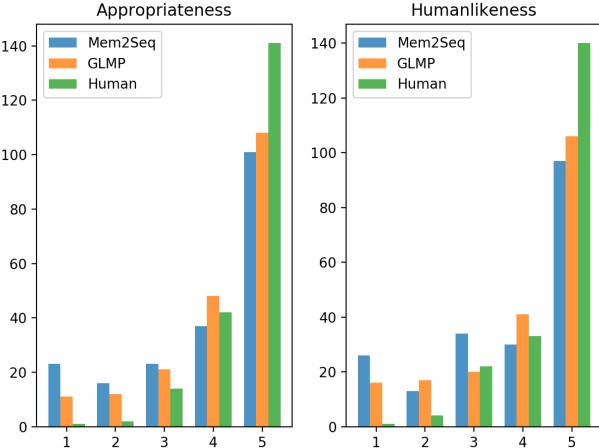

Figure 4: Appropriateness and human-likeness scores according to 200 dialogue scenarios.

Appropriateness
5: Correct grammar, correct logic, correct dialogue flow, and correct entity provided

4: Correct dialogue flow, logic and grammar but has slightly mistakes in entity provided
3: Noticeable mistakes about grammar or logic or entity provided but acceptable
2: Poor grammar, logic and entity provided
1: Wrong grammar, wrong logic, wrong dialogue flow, and wrong entity provided

Human-Likeness (Naturalness)
5: The utterance is 100% like what a person will say
4: The utterance is 75% like what a person will say
3: The utterance is 50% like what a person will say
2: The utterance is 25% like what a person will say
1: The utterance is 0% like what a person will say

## B    ERROR ANALYSIS

For bAbI dialogues, the mistakes are mainly from task 3, which is recommending restaurants based on their rating from high to low. We found that sometimes the system will keep sending those restaurants with the higher score even if the user rejected them in the previous turns. On the other hand, SMD is more challenging for response generation. First, we found that the model makes mistakes when the KB has several options corresponding to the user intention. For example, once the user has more than one doctor appointment in the table, the model can barely recognize. In addition, since we do not include the domain specific and user intention supervision, wrong delexicalized responses may be generated, which results in an incorrect entity copy. Lastly, we found that the copied entities may not be matched to the generated sketch tags. For example, an address tag may result in a distance entity copy. We leave the space of improvement to future works.

## C    ADDITIONAL DISCUSSION

One of the reviewers suggested us to compare our work to some existing dialogue framework such as PyDial [5]. To the best of our knowledge, in the PyDial framework, it requires to have the dialogue acts labels for the NLU module and the belief states labels for the belief tracker module. The biggest challenge is we do not have such labels in the SMD and bAbI datasets. Moreover, the semi tracker in PyDial is rule-based, which need to re-write rules whenever it encounters a new domain or new datasets. Even its dialogue management module could be a learning solution like policy networks, the input of the policy network is still the hand-crafted state features and labels. Therefore, without the rules and labels predefined in the NLU and belief tracker modules, PyDial could not learn a good policy network.

Truly speaking, based on the data we have (not very big size) and the current state-of-the-art machine learning algorithms and models, we believe that a well and carefully constructed task-oriented dialogue system using PyDial in a known domain using human rules (in NLU and Belief Tracker) with policy networks may outperform the end-to-end systems (more robust). However, in this paper, without additional human labels and human rules, we want to explore the potential and the advantage of end-to-end systems. Besides easy to train, for multi-domain cases, or even zero-shot domain cases, we believe end-to-end approaches will have better adaptability compared to any rule-based systems.

---

[5]http://www.camdial.org/pydial/

# D VISUALIZATION

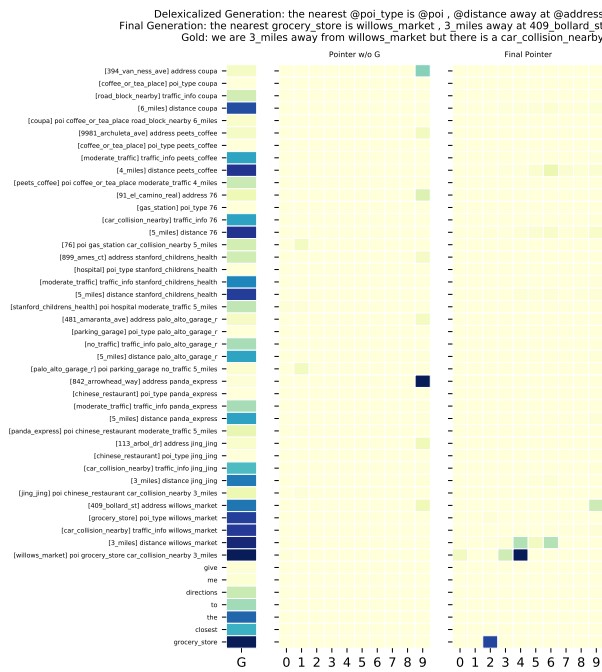

Figure 5: Memory attention visualization from the SMD navigation domain.

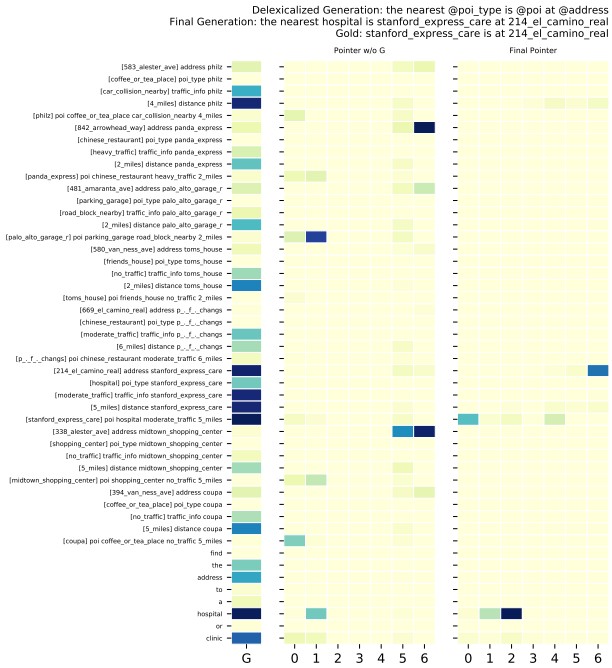

Figure 6: Memory attention visualization from the SMD navigation domain.

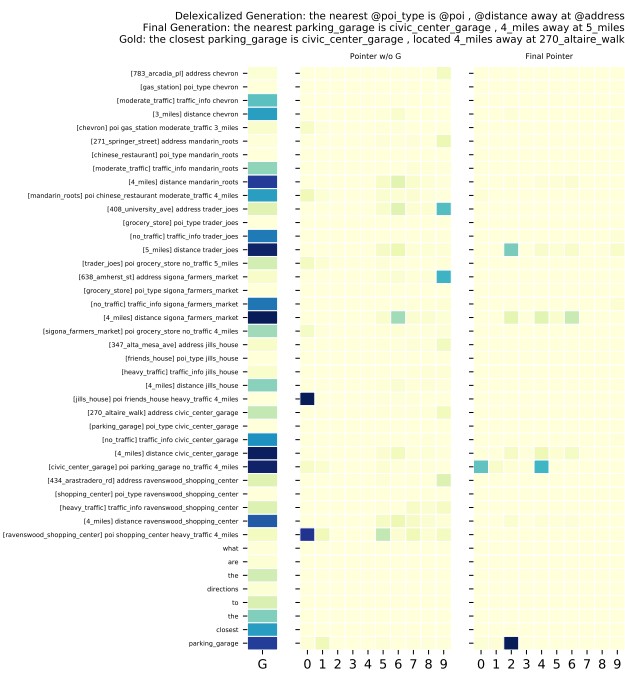

Figure 7: Memory attention visualization from the SMD navigation domain.

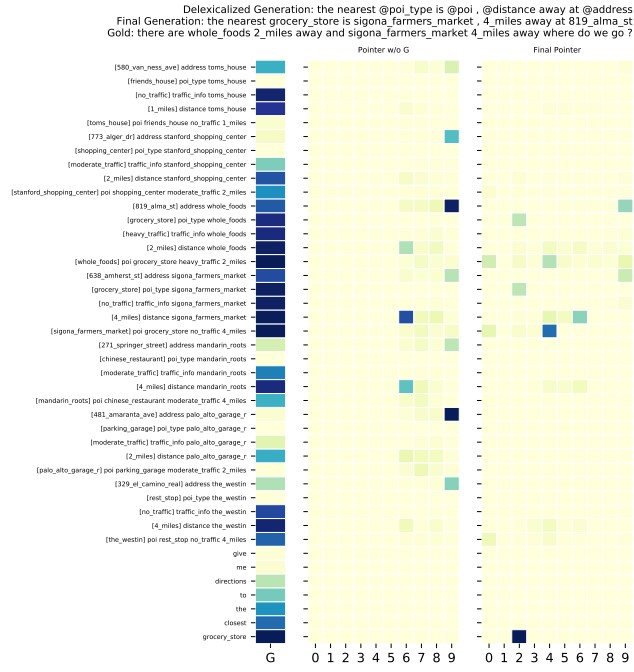

Figure 8: Memory attention visualization from the SMD navigation domain.

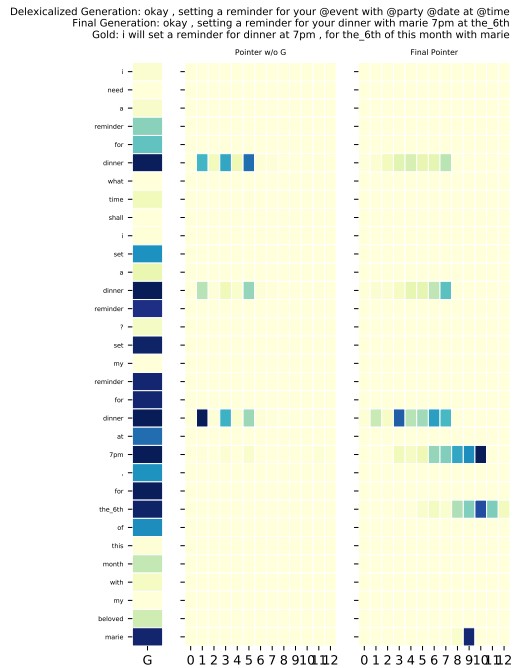

Figure 9: Memory attention visualization from the SMD schedule domain.

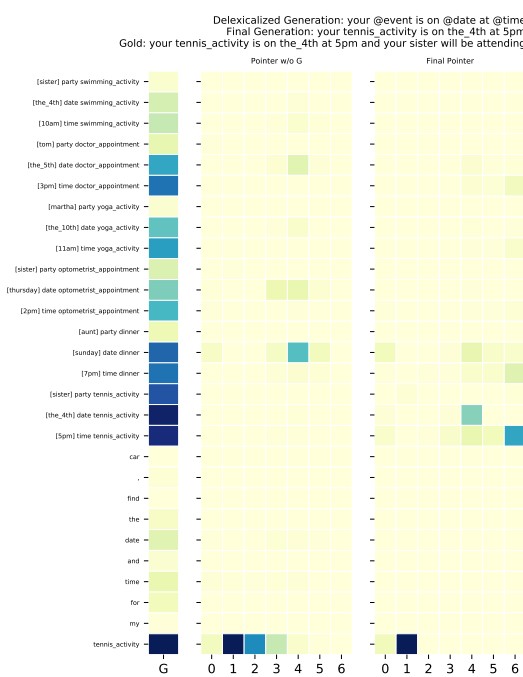

Figure 10: Memory attention visualization from the SMD schedule domain.

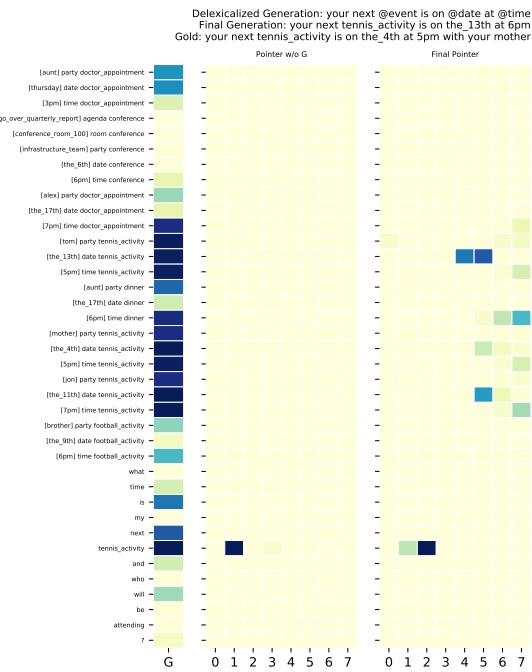

Figure 11: Memory attention visualization from the SMD schedule domain.

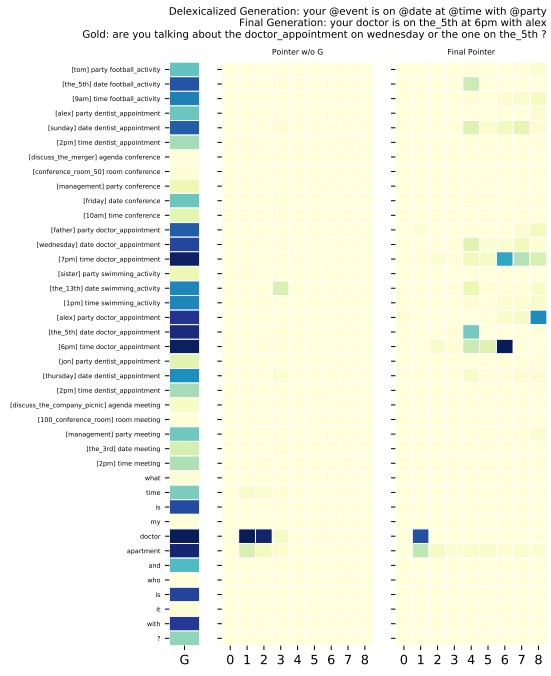

Figure 12: Memory attention visualization from the SMD schedule domain.

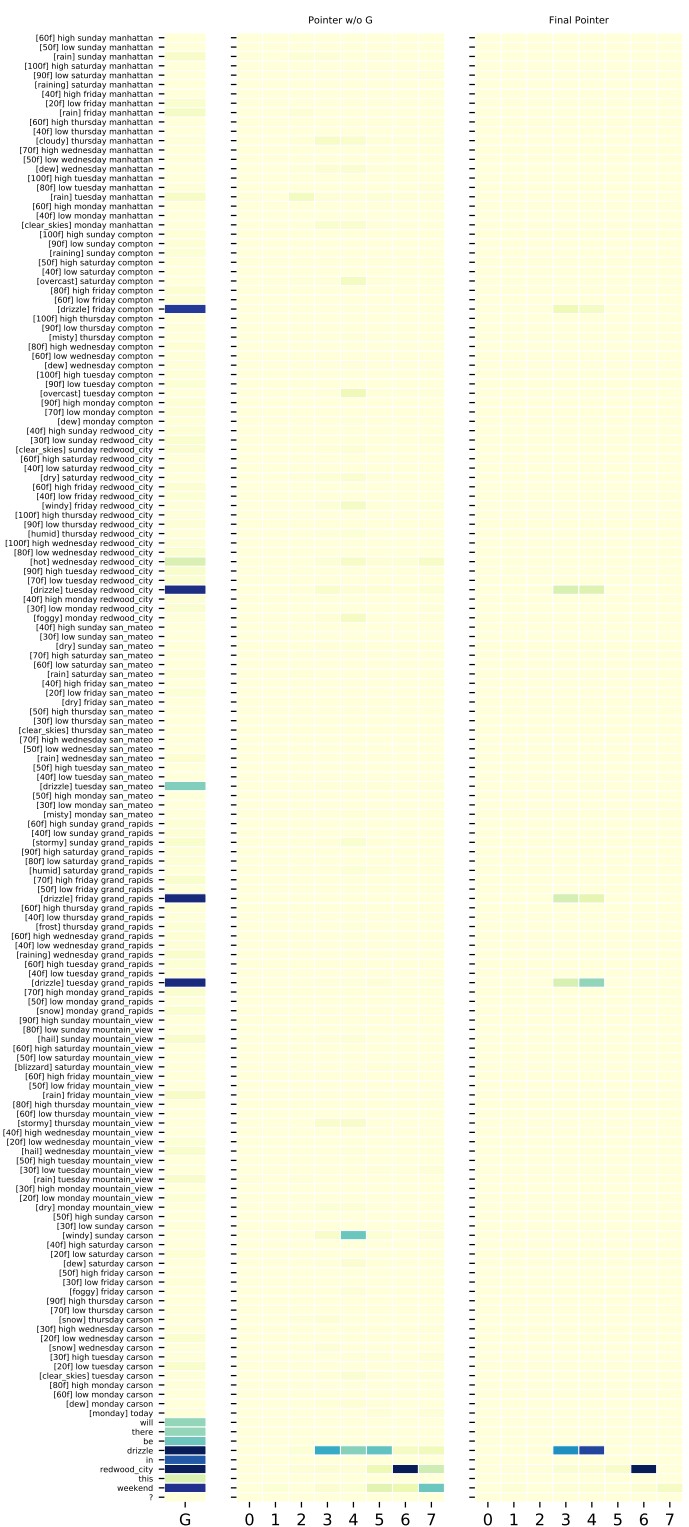

Figure 13: Memory attention visualization from the SMD weather domain.

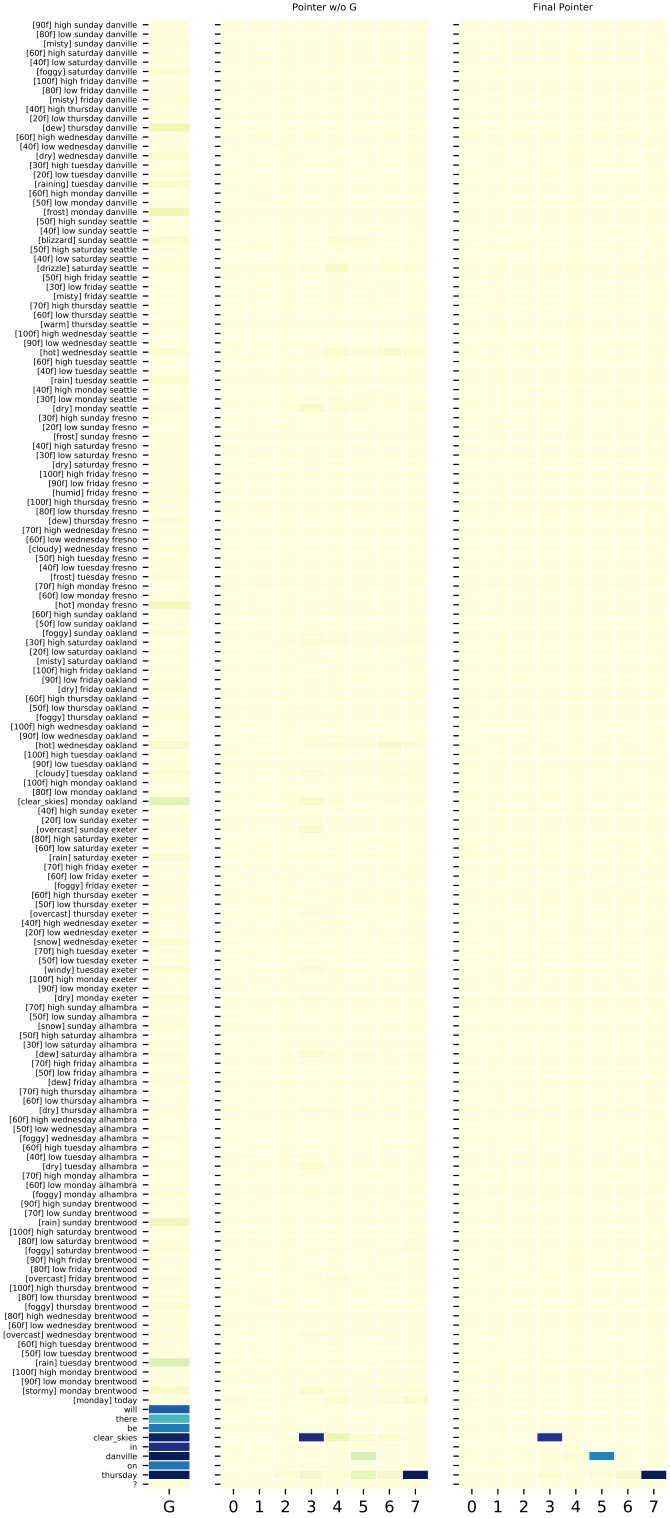

Figure 14: Memory attention visualization from the SMD weather domain.

