# OpenReview forum: "Global-to-local Memory Pointer Networks for Task-Oriented Dialogue"
_ICLR.cc/2019/Conference_

### Official Review · AnonReviewer1 · 2018-10-14
**Expect more experiments**

**Rating:** 5
**Confidence:** 3

**Review:**

This paper puts forward a new global+local memory pointer network to tackle task-oriented dialogue problem.

The idea of introducing global memory is novel and experimental results show its effectiveness to encode external knowledge in most cases.

Here're some comments:
1. In global memory pointer, the users employ non-normalized probability (non-softmax). What is the difference in performance if one uses softmax?

2. In (11), there's no linear weights. Will higher weights in global/local help?

3. As pointed out in ablation study, it's weird that in task5 global memory pointer does not help.

4. The main competitor of this algorithm is mem2seq. While mem2seq includes DSTC2 and In-car Assistant, and especially in-car assistant provides the first example dialogue, why does the paper not include expeirments on these two datasets?

---

> ### Author Response · Authors · 2018-11-11
> **Re: Reviewer1**
>
> Thank you for your review and feedback.  The question which you mentioned, the replies are as followed :
>
> 1. In global memory pointer, the users employ non-normalized probability (non-softmax). What is the difference in performance if one uses softmax?
> Reply:
> Sorry that we did not make it clear. We treat the training of global memory pointer as a multi-label learning problem, instead of a multi-class classification problem. For example, if the system generates a response like “Starbucks is 4_miles away”, both “Starbucks” and “4_miles” are model’s outputs.
>
> 2. In (11), there's no linear weights. Will higher weights in global/local help?
> Reply:
> In our experiments, we tried to add the linear weights in (11) as hyper-parameters, the results are similar.
>
> 3. As pointed out in ablation study, it's weird that in task5 global memory pointer does not help.
> Reply:
> Note that our ablation study on bAbI dialogue is on its OOV setting, which could be considered as a zero-shot scenario. Therefore, we observe a performance tradeoff between the normal setting and the OOV setting, that is, if we increase the dropout ratio, the accuracy in normal setting may decrease and the accuracy in OOV setting may increase. Therefore, after we rerun the ablation study on task5 OOV with the same dropout rate, the performance difference between with and without global memory pointer is very little. We will include this explanation in the paper.
>
> 4. The main competitor of this algorithm is mem2seq. While mem2seq includes DSTC2 and In-car Assistant, and especially in-car assistant provides the first example dialogue, why does the paper not include expeirments on these two datasets?
> Reply:
> Sorry that we did not make it clear. The “SMD” dataset in our experiment is exactly the same as the “In-Car Assistant” dataset in the Mem2Seq paper (different naming), both came from the paper Eric et al, 2017. Therefore, the results are comparable. Second, We did not include the DSTC2 in our paper is because it is a “human-machine” dataset which is originally designed as a DST task, not a response generation task. That dataset has many noisy system responses as well. We take one of the dialogues as an example, one can observe that the system responses are not properly collected.
> ...
> User: <SILENCE>
> System: the_lucky_star is a nice place in the south of town and the prices are cheap
> User: what else is there
> System: the_lucky_star is a great restaurant
> User: anything else
> System: The post code of the_lucky_star is the_lucky_star_post_code
> ...

---

### Official Review · AnonReviewer3 · 2018-10-29
**End-to-end task oriented system: An encoder-decoder approach with a shared external knowledge base**

**Rating:** 8
**Confidence:** 2

**Review:**

This is, in general, a well-written paper with extensive experimentation.

The authors tried to describe their architecture both with equations as well as graphically. However, I would like to mention the following:

In Section 2.1 I am not sure all the symbols are clearly defined. For example, I could not locate the definitions of n, l etc. Even if they are easy to assume, I am fond of appropriate definitions. Also, I suspect that some symbols, like n, are not used consistently across the manuscript.

I am also confused about the loss function. Which loss function is used when?

I am missing one more figure. From Fig 2 it's not so straightforward to see how the encoder/decoder along with the shared KB work at the same time (i.e. not independently)

In Section 2.3, it's not clear to me how the expected output word will be picked up from the local memory pointer. Same goes for the entity table.

How can you guarantee that that position n+l+1 is a null token?

What was the initial query vector and how did you initialise that? Did different initialisations had any effect on performance?

If you can please provide an example of a memory position.

Also, i would like to see a description of how the OOV tasks are handled.

Finally, your method is a NN end-to-end one and I was wondering how do you compare not with other end-to-end approaches, but with a traditional approach, such as pydial?


And some minor suggestions:

Not all the abbreviations are defined. For example QRN, GMN, KVR. It would also be nice to have the references of the respective methods included in the Tables or their captions.

Parts of Figs. 1&2 are pixelised. It would be nice to have everything vectorised.

 I would prefer to see the training details (in fact, I would even be favorable of having more of those) in the main body of the manuscript, rather than in the appendix.

There are some minor typos, such as "our approach that utilizing the recurrent" or "in each datasets"

---

> ### Author Response · Authors · 2018-11-11
> **Re: Reviewer3**
>
> Thank you for your review and feedback. The question which you mentioned, the replies are as followed :
>
> 1. In Section 2.1 I am not sure all the symbols are clearly defined.
> Reply:
> We will make the definitions more appropriate and consistent.
>
> 2. I am also confused about the loss function. Which loss function is used when?
> Reply:
> Our model has three loss functions: Loss_g for global memory pointer, Loss_v for sketch response generation and Loss_l for local memory pointer. During training, they are summed and optimized simultaneously.
>
> 3. I am missing one more figure. From Fig 2 it's not so straightforward to see how the encoder/decoder along with the shared KB work at the same time (i.e. not independently)
> Reply:
> As shown in the block diagram Fig 1(a), first, the global memory encoder encodes dialogue history and writes its hidden states into the external knowledge. Then the last hidden state is used to read the external knowledge and generate the global memory pointer at the same time. Later during the decoding stage, the local memory decoder generates sketch responses. Then the global memory pointer and the sketch RNN hidden state are passed to the external knowledge, which returns the local memory pointer that can copy plain text to replace the sketch tags and obtain the final system response.
>
> 4. In Section 2.3, it's not clear to me how the expected output word will be picked up from the local memory pointer. Same goes for the entity table.
> Reply:
> Sorry that we did not make it clear. As the visualization in Fig 3, the right column is the local memory pointers for time step 0 to 3. For example, in step 3, when our sketch RNN generated tags such as“@address”, the word will be picked out from the learned local memory pointer, which points to the memory node “[783_arcadia_pl] address chevron”. Therefore, we took the Object word “783_arcadia_pl” out as the real address. Otherwise, the output word is generated from the vocabulary
>
> 5. How can you guarantee that that position n+l+1 is a null token?
> Reply:
> We manually assign token of “n+l+1” position to be “NULL” during preprocessing.
>
> 6. What was the initial query vector and how did you initialise that? Did different initialisations had any effect on performance?
> Reply:
> The query vector is the vector to query the external knowledge. In the encoder, the query vector is the last hidden state of context RNN. In the decoder, the query vectors are the hidden states of the sketch RNN.
>
> 7. If you can please provide an example of a memory position.
> Reply:
> The example of memory position is shown in the left part of Fig 3, as you can see, our external knowledge includes the kB and the dialogue history.
>
> 8. Also, i would like to see a description of how the OOV tasks are handled.
> Reply:
> Sorry that we did not make it clear. In Sec 2.2, we explain that our model can mitigate the OOV problem is because we use the context RNN hidden states as the global contextual representation, and feed into the external knowledge. Therefore, the embedding of each token includes its RNN hidden state, including embeddings of OOV tokens.
>
> 9. Finally, your method is a NN end-to-end one and I was wondering how do you compare not with other end-to-end approaches, but with a traditional approach, such as pydial?
> Reply:
> We mainly followed previous works to compare end-to-end models without human feature engineer efforts. In Table 3, results of the rule-based system from the Eric et al., 2017 are reported, we can observe the improvement over the traditional pipeline solution on the SMD human-human dialogue dataset.

---

> > ### Comment · AnonReviewer3 · 2018-11-22
> > **Thanks - just one point**
> >
> > For 9. I would still be interested (if possible and straight-forward) to see how you compare to pydial (that is not an encoder-decoder approach), since pydial policy manager is also NN (and not rule-based) as the Eric et al. 2017.

---

> > > ### Author Response · Authors · 2018-11-26
> > > **Re: Reviewer3**
> > >
> > > Yes, we agree with you that it will be interesting to have a comparison of the end-to-end systems with the modularized systems. However, please let us show some difficulties to design a system like that using pydial in the SMD and bAbI datasets we used in our paper:
> > >
> > > To the best of our knowledge, in the pydial framework, it requires to have the dialogue act’s labels for the NLU module and the belief states’ labels for the belief tracker module. The biggest challenge here is we do not have such labels in the SMD and bAbI datasets we used. Moreover, the semi tracker in pydial is rule-based (ex: self.slot_vocab["pricerange"] = "(price|cost)(\ ?range)*"), which need to re-write rules whenever it encounters a new domain or new datasets. Even its dialogue management module could be a learning solution like policy networks, the input of the policy network is still the hand-crafted state features and labels. Therefore, without the rules and labels predefined in the NLU and belief tracker modules, pydial couldn’t learn a good policy network.
> > >
> > > Lastly, for now, based on the data we have (not very big size) and the current SOTA machine learning algorithms and models, we believe that a well and carefully constructed task-oriented dialogue system (like pydial) in a known domain using human rules (in NLU and Belief Tracker) with policy networks may outperform the end-to-end systems. However, in this paper, without additional human labels and human rules, we want to explore the potential and the advantage of end-to-end systems. Besides easy to train, for multi-domain cases, or even zero-shot domain cases, we believe end-to-end approaches will have better adaptability compared to any rule-based systems. We will include this discussion in our paper.
> > >
> > > Thank you again for your feedback and we really appreciate it.

---

> > > > ### Comment · AnonReviewer3 · 2018-11-26
> > > > **Covered**
> > > >
> > > > Many thanks for the most detailed reply. It was most enlightening. Yes please, do add that to the discussion. I believe many people in the field would be interested in your point of view. Many thanks again!

---

### Official Review · AnonReviewer2 · 2018-11-03
**nicely motivated architecture and thorough evaluation, aimed at an interesting and difficult task**

**Rating:** 8
**Confidence:** 2

**Review:**

The paper presents a new model for reading and writing memory in the context of task-oriented dialogue. The model contains three main components: an encoder, a decoder, and an external KB. The external KB is in the format of an SVO triple store. The encoder encodes the dialogue history and, in doing so, writes its hidden states to memory and generates a "global memory pointer" as its last hidden state. The decoder takes as input the global memory pointer, the encoded dialogue state history, and the external KB and then generates a response using a two-step process in which it 1) generates a template response using tags to designate slots that need filling and 2) looks up the correct filler for each slot using the template+global memory pointer as a query. The authors evaluate the model on a simulated dialogue dataset (bAbI) and on a human-human dataset (Stanford Multi-domain Dialogue or SMD) as well as in a human eval. They show substantial improvements over existing models on SMD (the more interesting of the datasets) in terms of entity F1--i.e. the number of correctly-generated entities in the response. They also show improvement on bAbI specifically on cases involving OOVs. On the human evaluation, they show improvements in terms of both "appropriateness" and "human-likeness".

Overall, I think this is a nice and well-motivated model. I very much appreciate the thoroughness of the evaluation (two different datasets, plus a human evaluation). The level of analysis of the model was also good, although there (inevitably) could have been more. Since it is such a complex model, I would have liked to see more thorough ablations or at least better descriptions of the baselines, in order to better understand which specific pieces of the model yield which types of gains. A few particular questions below:

- You describe the auxiliary loss on the global pointer, and mention an ablation study that show that this improves performance. Maybe I am overlooking something, but I cannot find this ablation in the paper or appendix. It would be nice to see how large the effect is.
- Following on the above, why no similar auxiliary losses on additional components, e.g. the template generation? Were these tried and deemed unnecessary or vice-versa (i.e. the default was no auxiliary loss and they were only added when needed)? Either way, it would be nice to better communicate the experiments/intuitions that motivated the particular architecture you arrived at.
- I really appreciate that you run a human eval. But why not have humans evaluate objective "correctness" as well? It seems trivial to ask people to say whether or not the answer is correct/communicates the same information as the gold.

---

> ### Author Response · Authors · 2018-11-11
> **Re: Reviewer2**
>
> Thank you for your review and feedback. The question which you mentioned, the replies are as followed :
>
> 1. You describe the auxiliary loss on the global pointer, and mention an ablation study that show that this improves performance. Maybe I am overlooking something, but I cannot find this ablation in the paper or appendix. It would be nice to see how large the effect is.
> Reply:
> The ablation study of our global memory pointer G is in Table 4, the GLMP w/o G. For SMD dataset, without G gave us around 8.3% additional loss.
>
> 2. Following on the above, why no similar auxiliary losses on additional components, e.g. the template generation? Were these tried and deemed unnecessary or vice-versa (i.e. the default was no auxiliary loss and they were only added when needed)?
> Reply:
> Our model has three loss functions, Loss_g for global memory pointer, Loss_v for sketch response generation and Loss_l for local memory pointer. The template generation loss you mentioned is included as Loss_v, which is a standard cross-entropy loss.
>
> 3. I really appreciate that you run a human eval. But why not have humans evaluate objective "correctness" as well?
> Reply:
> In our evaluation setting, we combine the correctness and the appropriateness, as the criteria we mentioned in the appendix A.3.

---

### Public Comment · (anonymous) · 2018-11-16
**Interesting Work, Need Clarity on Experiements**

This paper builds upon Mem2Seq (Madotto et al 2018) to incorporate large external KBs for task-oriented dialogs. To the best of my understanding, the innovations over Mem2Seq are: (1) use of context RNN (instead of just last utterance) as the query in MN encoder, (2) addition of the hidden state of context RNN to the dialog memory (equation 3), (3) the use of global memory pointer 4) additional loss components (Loss_g and Loss_l in equation 11) and (5) two step decoding using sketch tags

To me the biggest pro of the paper is its impressive result on SMD corpus. I appreciate the authors performing a human evaluation of the generated response for SMD.  However, I am quite concerned about two main issues. The first issue is in experimental rigor, and second is in dataset preparation, which is also related to experimental rigor but also exposes certain model weaknesses. I elaborate below.

Experimental Rigor
This paper uses "entity type" information in Local Memory Decoder (Section 2.3), but compares against all previous work that does NOT use this information. This makes the comparison not sound. In fact, the original Memory Net paper (Bordes & Weston 2017) and many extensions performed two sets of experiments, one vanilla and one with a "match" feature, which had access to the entity type information. This paper compares against all previous papers in the settings that do NOT use this "match" feature. So, it is not clear, whether the improvement is coming from the specific changes in the model, or just by using additional information at training time. For example, QRN paper (Seo et al 2017) with Match feature reports an average OOV error (across 5 bAbI tasks) of 2.3%, which is fairly close the reported results in Table 2 in this paper. In my opinion, this careful experimental comparison is essential before having a clear assessment of this paper.

Dataset Preparation
This is a comment on this paper and also the Mem2Seq paper. Both these seem to have CHANGED the original training/test datasets to suit their model. In particular, all KB tuples in bAbI follow the format (restaurant_name, relation, value of relation), e.g., (olive_garden, rating, 5), however, Mem2Seq had reversed just the rating-relation tuples (5, rating, olive_garden), because its model allowed it to copy only from the object location, and it needed to copy restaurant name in the dialog. A similar example can be seen in this paper Figure 3 (row 5) where an ARTIFICIAL tuple (chinese_restaurant no_traffic 6_miles, poi, tai_pan) has been added to the SMD KB. Notice that values for different relations (namely poi_type, traffic_info, and distance) for the entity tai_pan have been concatenated in this artificial tuple at subject location and the entity name appears as object. This is just so that it can be copied by the model. Such tuples don't usually exist in normal KBs.

I believe that this changing of datasets makes its comparisons with other models unfair. Even if other models were re-trained with this new modified dataset, it is still a severe limitation, because in practice, such tuples may not be found in the KB, and in such situations this model (and Mem2Seq) will not perform well.


Questions to authors
1) how does your model compare with "+match" extension of previous models?
2) Say we are in a more reasonable setting where we are not given entity type information, but say we are given whether it is a KB entity or not. How well will your model perform then?
3) Suppose we remove the reverse relations in bAbI and the concatenated poi relations in SMD. How well will your model perform then?

---

> ### Author Response · Authors · 2018-11-17
> **Thank you for your feedback and your clear summary of our contribution**
>
> Please let me reply to you below:
>
> First, in our experiment, we did not add the “entity type” information into the word representation, which is same as the previous works such as Mem2Seq, MemNN, QRN, etc. Therefore, the comparison is fair. The step we did related to entity type was the sketch response preprocessing, based on the provided entity table (or the NER if the table is not provided), we can obtain our gold sketch responses for training. The local memory pointer is then learned to copy words to replace the generated sketch tags. Note that all the word-level representations in the external knowledge are not included the “type embedding”.
>
> Second, yes we followed the same preprocessing as in the Mem2Seq paper to represent our KB tuples. If you look into the original KB in the SMD dataset, it is not represented as the triplet format. Therefore, there are many different ways to represent the KB information, some may use flat KB as we did, some may use the hierarchical one, or even the input the table-like KB. Although it might not be the most effective one, we choose this preprocessing strategy, so does the previous works, is because it is simple and fast. There are some related works have tried different ways to represent KB information, but it may need additional attention calculation for entity copying. The comparison between these KB structures are interesting and could be our future works.
>
> Thank you again for your interest in our work.

---

> > ### Public Comment · (anonymous) · 2018-11-23
> > **a few concerns**
> >
> > Thank you for your reply, but some of my concerns are still unclear.
> >
> > Entity Type Information:
> > Thank you for clearly stating that GLMP uses entity type information to create sketch responses. You have mentioned GLMP did not add the “entity type” information into the word representation, but neither did the existing approaches that use match feature (QRN, MemNN and Gated MemNN).
> >
> > At a high level, there exists one set of approaches that use entity type information (in the model), and GLMP also uses the same information (in a different way). However, you have only compared against the models that do not use this information at all. This does not feel like a fair comparison.
> >
> > Dataset Preparation:
> > Thank you for explaining the preprocessing in the SMD dataset, we felt it was not consistent with the explanation in section (2.1) which states "each element bi ∈ B is represented in the triplet format as (Subject, Relation, Object) structure". Any comments about the preprocessing in the bAbI dataset? The reason for posing the question (Q3) in the initial comment is that we strongly feel that removing the preprocessing would considerably reduce the accuracy as well as task completion rate for T3-OOV and T5-OOV.
> >
> > We would appreciate if you can answer the specific questions we raised. Your silence, suggests that the answers are likely,
> > (Q1) very close performance
> > (Q2) not very good
> > (Q3) would not work very well

---

> > > ### Author Response · Authors · 2018-11-26
> > > **Thank you for raising your concerns**
> > >
> > > Please let us reply you as below:
> > >
> > > 1) The entity type information we mentioned is the slot type representation (embedding) for each slot. For example, to the best of our knowledge, in the MemNN for bAbi dialogue, they have 7 special words (embeddings) for 7 different slot types that they then added to all the words that are related to them. In this way, for example, the representation of “Paris” can include the information of “location”.
> > >
> > > In our case, our model does not have the explicit information in the “Paris embedding” that it is a “location”. Please note that when we encode the dialogue history, we used the plain text input, that is, the “Paris” embedding does not include the “location” type embedding. Even if an OOV word comes into place, our model did not add the type information as the others did, instead, we added the hidden states of the RNN encoder. The sketch response is only used while "decoding", not "encoding". During the encoding stage, all the input are plain texts, not the sketch sentences. Hope this makes it clear about your question one and two.
> > >
> > > 2) We total understood your concern about the KB. Please note that for each “node”, we summed up the embeddings of (Subject, Relation, Object), then we assume that every time the “node” is pointed to, we copy the Object word out (it’s our own rule we defined). Therefore, there is no constraint that what needs to be a subject and what needs to be an object. The only thing important in our task is we need to be able to copy every entity that may exist in a response. Thus, we need at least a node that we can copy the “name of entity” out, for example, the restaurant name. That is, either we decide to copy the Subject for the entity names, or we just simply represent the name of the entity as an Object in one node. This is a matter of design. We do this is because it is easy for us to maintain our code. Therefore, hope this explains your question three.
> > >
> > > In addition, as we mentioned in the last post, there are many different ways to represent the KB information, some may use flat KB as we did (ex: mem2seq), some may use the hierarchical one. Although flat KB might not be the most effective one (because the hierarchical one is easier for machine to reason KB, the nodes are assumed to connect by the entity names), we choose this preprocessing strategy and left the ability of connecting the nodes to our system, so does some previous works, is because it is simple and fast. The comparison between these two could be an interesting future work.
> > >
> > > 3) Lastly, we will release our code if our work is published. If you have any further question about the preprocessing or model architecture, etc, we hope that can make you more clear.
> > >
> > > Thank you again for your interests in our work. Very happy to hear that.

---

### Meta-Review · Area_Chair1 · 2018-11-06
**novel architecture for task oriented dialogue systems**

**Confidence:** 4
**Recommendation:** Accept (Poster)

**Metareview:**

Interesting paper applying memory networks that encode external knowledge (represented in the form of triples) and conversation context for task oriented dialogues. Experiments demonstrate improvements over the state of the art on two public datasets.
Notation and presentation in the first version of the paper were not very clear, hence many question and answers were exchanged during the reviews.